# Permeabilities of CO_2_, H_2_S and CH_4_ through Choline-Based Ionic Liquids: Atomistic-Scale Simulations

**DOI:** 10.3390/molecules24102014

**Published:** 2019-05-27

**Authors:** Abdukarem Amhamed, Mert Atilhan, Golibjon Berdiyorov

**Affiliations:** 1Qatar Environment and Energy Research Institute, Hamad Bin Khalifa University, Doha 34110, Qatar; gberdiyorov@hbku.edu.qa; 2Department of Chemical Engineering, Texas A&M University at Qatar, Doha 23874, Qatar; mert.atilhan@gmail.com; 3Gas and Fuels Research Center, Texas A&M University, College Station, TX 77843, USA

**Keywords:** ionic liquid, gas separation, molecular dynamics

## Abstract

Molecular dynamics simulations are used to study the transport of CO${}_{2}$, H${}_{2}$S and CH${}_{4}$ molecules across environmentally friendly choline-benzoate and choline-lactate ionic liquids (ILs). The permeability coefficients of the considered molecules are calculated using the free energy and diffusion rate profiles. Both systems show the largest resistance to CH${}_{4}$, whereas more than 5 orders of magnitude larger permeability coefficients are obtained for the other two gas molecules. The CO${}_{2}$/CH${}_{4}$ and H${}_{2}$S/CH${}_{4}$ selectivity was estimated to be more than 10${}^{4}$ and 10${}^{5}$, respectively. These results indicate the great potential of the considered ILs for greenhouse gas control.

## 1. Introduction

Recent global climate changes necessitate the capture and storage of greenhouse gases (especially carbon dioxide, CO${}_{2}$) for their efficient mitigation [1,2,3,4,5]. Recently, pre-combustion carbon capture using ionic liquids (ILs) has received a lot of interest due to small degradation, high CO${}_{2}$ absorption rate and environmental issues as compared to post-combustion carbon capture methods (see Reference [6,7,8,9,10,11] for reviews). As an example, the consumption energy in conventional systems applying monoethanolamine is about 4.2 GJ/tone CO${}_{2}$ [12], whereas in piperazine and other scrubbing systems, the regeneration heat consumption is 2.6 MJ/CO${}_{2}$ [13,14,15]. ILs are considered as promising solvents to enhance CO${}_{2}$ capture efficiency and to reduce the cost [16]. In addition, the regenerator tower is the most energy consumption unit of the CO${}_{2}$ removal system [3,4]. In this respect, the ILs have a potential for Acid Gas Removal process due to their high thermal and chemical stability [17], negligible vapour pressures [18], low flammability and toxicity [19,20,21] and low cost carbon regeneration [7,22]. ILs also present the possibility of creating task-specific materials with desired chemical, physical and mechanical properties [23,24]. These properties make ILs a good alternative candidate to environmentally volatile organic molecular solvents for carbon capture and separation applications. In general, ILs consist of large organic cations and organic/inorganic anions with low melting points (<100 ${}^{\circ }$C) and exhibit unique physical and chemical properties such as good thermal properties, improved structural stability, non flammability, and more interestingly very low vapor pressure [25]. Large number of potential ILs can be created using different combinations of anions and cations [26]. However, the development of efficient ILs for carbon capture and separation applications requires a fundamental understanding of their properties. Molecular modeling has proven to be an effective tool in describing the relationship between the structure of ILs and their gas absorption properties [27,28] (see Reference [29] for a review).

Among the many families of ILs, choline [CH] cation-based ILs have attracted a lot of interest in recent years due to their improved biodegradability and low-cost syntheses [30,31,32]. CO${}_{2}$ absorption in choline-based ILs was recently studied by Aparicio et al. [33,34,35,36] using first-principles density functional theory (DFT) calculations and force-field-based molecular dynamics (MD) simulations. The simulations showed the importance of anions in choline-based ILs. For example, aromatic ions (such as benzoate [BE]) have considerably smaller molecular mobilities as compared to nonaromatic ions (such as lactate [LAC]). Therefore, a fundamental understanding of the relationship between the molecular structure and the gas absorption properties of ILs is required for the design of efficient ILs for carbon capture and separation applications.

In this work, we perform empirical force-field based MD simulations to study the permeability of CO${}_{2}$, H${}_{2}$S and CH${}_{4}$ molecules through choline-benzoate ([CH][BE], see Figure 1a) and choline-lactate ([CH][LAC], see Figure 1b), which are shown to be cost-effective and environmentally friendly materials (so called biomaterials) [30,31]. The force fields are created on the automated topology builder (ATB) repository using B3LYP/6-31G* level of theory + Hessian. The permeability coefficients, which are one of the main parameters defining the gas absorption capabilities of ILs [37,38,39], are obtained using the generalized Langevin equation (GLE) method, by calculating the free energy and diffusion rate profiles for the considered solutes.

## 2. Computational Method

Due to difficulties of atomic level probing of ILs experimentally, computer simulations became an unprecedented tool to get information on atomistic scale. In this regard, force-field based MD simulations present a particular interest due to the possibility of modelling larger systems and longer reaction times. In this work, the considered systems are described using the GROMOS 53A6 all atom force field with the parameter set obtained from the ATB repository. The corresponding ATB models are 9873, 8515 and 10159, respectively for choline, benzoate and lactate [40]. The force field parameters are obtained from DFT calculations using the hybrid B3LYP/6-31G* exchange-correlation functional (see Reference [41] for more details about the ATB calculation procedure). Ions in both ILs have full charges (±1|e|). As for the non-bonded interactions, a 1.0 nm cut-off is applied to account for the van der Waals interactions. The electrostatic interactions, on the other hand, are calculated using the PME-method [42], using a 1.0 nm cut-off for the real-space interactions in combination with a 0.12 nm spaced-grid for the reciprocal-space interactions and a fourth-order B-spline interpolation. After geometry optimization of the individual molecules (see Figure 1c), we have constructed our model systems, using the Packmol package [43], consisting of 200 choline and 200 lactate molecules ([CH][LAC] system) and 200 choline and 200 benzoate molecules ([CH][BE] system). We have performed geometry optimization for the initial configurations using the steepest descent algorithm. The systems are further equilibrated at 300 K for 50 ns using an isothermal isobaric ensemble with a Nosé-Hoover thermostat [44] and the semi-isotropic Parinello-Rahman barostat [45]. The damping constants for temperature and pressure are 200 fs and 1 ps, respectively, and the time step was 2 fs. The applied pressure was 1 atmosphere with a compressibility constant of 4.5 × 10${}^{-5}$ bar${}^{-1}$. Periodic boundary conditions were applied in all Cartesian directions. This thermalization resulted in the lattice parameters 3.77 × 3.77 × 3.77 nm${}^{3}$ and 3.86 × 3.86 × 3.86 nm${}^{3}$, respectively for the [CH][LAC] and [CH][BE] systems (see Supplemental material). Subsequently, the structures were allowed to expand in the z-direction for the next 100 ns simulations. The simulations are conducted using a canonical (NVT) ensemble with the Nose-Hoover thermostat with the lattice parameter of 14 nm in the z-direction for both systems. The purpose was to create a liquid-gas interface to investigate the permeability of CH${}_{4}$, CO${}_{2}$ and H${}_{2}$S molecules through these ILs. Corresponding ATB models are 15610 for CH${}_{4}$, 6108 for CO${}_{2}$ and 3613 for H${}_{2}$S. The last 5 ns of the NVT runs were used to extract five structures (with 1 ns interval) that are used further in umbrella sampling (US) simulations. 6 US windows (separated by 10 Å, see dashed red horizontal lines in Figure 1a,b) are sampled during each simulation and 10 US calculations are performed to obtain a single free energy profile. The motion of CH${}_{4}$, CO${}_{2}$ and H${}_{2}$S molecules is restricted along the *z*-direction using a 2000 kJ/mol/nm force constant, whereas the molecules are allowed to move freely along the other two directions. The free energy profiles are constructed by using the weighted histogram analysis method [46] after 4 ns US MD simulations. The final energy profiles are obtained by averaging over 250 US simulations (i.e., by averaging over 25 individual energy profiles). The position-dependent diffusion coefficients of the considered solutes are calculated using the GLE method [47,48]. The results presented here are the average over 25 individual diffusion rate profiles. Using the free energy profiles and the diffusion coefficients D(z), we have calculated the permeability coefficient ($P_{m}$) using [48]
(1)$$\begin{array}{c} \frac{1}{P_{m}}=\int_{z1}^{z_{2}}\frac{e^{w\left(z\right)/k_{B}T}}{D(z)}dz, \end{array}$$
where w(z) is the potential of mean force and the integral is over an interval of *z* that spans the ILs. The diffusion coefficient of a solute in each z position is calculated using the position autocorrelation function $C_{z}\left(t\right)$ as
(2)$$\begin{array}{c} D\left(z_{i}={\langle z\rangle }_{i}\right)=\frac{\mathrm{var}{\left(z\right)}^{2}}{\int_{0}^{\infty }C_{z}\left(t\right)dt}. \end{array}$$

All simulations and analysis were performed using the GROMACS 5.1 package [49].

## 3. Results and Discussion

We started by calculating densities of the considered ILs without the gas molecules to validate our force fields by comparing them to the experimental data and previous simulation results. The dashed grey curves in Figure 2 show the density of the systems to highlight the liquid-vapor interface. Note that the calculated density of [CH][BE] is in good agreement with previous MD calculations [35] and experiments [30].

Next, we calculate free energy profiles for the transfer of the considered gas molecules across the gas-liquid interfaces of the considered ILs, which enables us to assess the equilibrium solvation behavior of the gas molecules. The transfer occurs along the normal to the interface along the *z*-direction. Figure 2a,b shows the excess free energy profiles ΔG of CH${}_{4}$, CO${}_{2}$ and H${}_{2}$S molecules in [CH][BE] (a) and [CH][LAC] (b) ILs. The free energy curve of CO${}_{2}$ in [CH][BE] shows local minima near the interface and a plateau-like maximum deeper inside the IL (red curve in Figure 2a). A similar free energy profile is obtained for H${}_{2}$S (blue curve in Figure 2a). The difference is in the value of ΔG, which is ≈20 kJ/mol smaller in the case of CO${}_{2}$. Similar free energy profiles of both molecules are obtained in [CH][LAC] (Figure 2b). Note that in both systems the absorption of both molecules is energetically favorable with more profound absorption of H${}_{2}$S molecules. The average values of the free energy in the [CH][BE] system are −7.84 kJ/mol and −22.21 kJ/mol, respectively for CO${}_{2}$ and H${}_{2}$S molecules. In the case of the [CH][LAC] system, these numbers are −4.73 kJ/mol and −21.35 kJ/mol. We believe that the larger free energy for H${}_{2}$S may originate from its permanent dipole moment, which does not exist in CO${}_{2}$.

The excess free energy profile of CH${}_{4}$ is completely different in both ILs: the solvation of these molecules is energetically unfavorable except at the liquid-vapor interface (black curves in Figure 2a,b). Moreover, ΔG of CH${}_{4}$ at the interface is much smaller than to the ones obtained for the other two molecules. The averaged values of the excess free energy of CH${}_{4}$ are 11.79 kJ/mol and 11.52 kJ/mol, respectively in the [CH][BE] and [CH][LAC] ILs.

Figure 2c,d show the diffusion coefficients of the considered molecules at various *z* positions, calculated with Equation (Equation 2), for the [CH][BE] (c) and [CH][LAC] (d) systems. It is again clear from these figures that the diffusion patterns of CO${}_{2}$ (red curves) and H${}_{2}$S (blue curves) molecules are similar. The highest diffusion coefficients are found at the interface, where the density of the ILs is the lowest, and smaller diffusion coefficients deeper inside the ILs, where the density becomes larger. The diffusion rate drops by more than a factor 2 when the molecule penetrates the systems. In both systems, the diffusion coefficient of CH${}_{4}$ is considerably larger than for the other two molecules, both at the interface and in the bulk (see black curves in Figure 2c,d). This indicates a smaller effective permeability of CH${}_{4}$. In addition, diffusion rate of methane drops more sharply when the molecule penetrates the ILs. The variations of the *z*-dependent diffusion coefficient is also larger in the case of CH${}_{4}$.

Note that in the present model, both cations and anions have “full” charges (±1|e|) in the present systems. It was shown in previous force field based MD simulations that such “full” charge description of the ionic liquids may underestimate the dynamic properties (e.g., self-diffusivity) of the ions in the system, despite accurate description of the static properties (e.g., density) of the materials [51,52]. Simple scaling of the ions charges were proposed to be a good fitting parameter to get better agreement with the experiment. However, we did not perform such scaling in the charge partitioning expecting the same level of slowing down the dynamics of the considered gas molecules, i.e., similar qualitative results are expected for the selectivity of the present ILs for the considered gas molecules.

It is well known that ILs are often characterized by very high viscosity due to, e.g., strong coulombic interactions. Therefore, much longer simulations times (in ns range) are required to reach the dynamic equilibrium in the system and to estimate the diffusion coefficients [53,54]. In order to reach the equilibrium in the considered systems, we further performed 50 ns MD simulations for single solute molecule randomly placed inside the ILs. Even such long simulations result in considerably fluctuations of the diffusion coefficients (see Figure 3). Therefore, in order to better estimate diffusion coefficient of the considered solutes in the bulk of the ILs, we have conducted statistical averaging over different initial configurations. For that purpose we have placed a single gas molecule in 20 different positions randomly inside the bulk. This will improve the statistics of determining the diffusion coefficients. To avoid artificial problems in the calculations due to the overlap between the atoms, we have used large enough intermolecular distance during the generation of the gas molecules using Packmol. In addition, we have conducted structural optimizations for every random generation of the gas molecules inside the system followed by NPT simulations for the relaxation of lattice parameters of the simulation box before running long NVT simulations with periodic boundary conditions. The averaged diffusion coefficients (calculated from the mean square displacements) for CH${}_{4}$, CO${}_{2}$ and H${}_{2}$S molecules are 11.71, 6.57 and 2.21 ($\times 10^{-9}$ m${}^{2}$/s), respectively, inside the [CH][BE] IL. For [CH][LAC] system we obtained the average values 4.66, 6.09 and 4.72 ($\times 10^{-9}$ m${}^{2}$/s), respectively for CH${}_{4}$, CO${}_{2}$ and H${}_{2}$S gas molecules. These values are in good agreements with the ones obtained in US simulations (see Figure 2c,d).

Since it is very challenging to directly measure the diffusion coefficients of small molecules within the ILs, we have computed the permeability coefficients of the considered molecules using the free energy and diffusion rate profiles. Following the previously reported literatures [55,56] we presented the permeability coefficient in m/s unit. The calculated permeability coefficients are given in Table 1. For CO${}_{2}$ we obtained values of $P_{CO_{2}}$ to be 2.25 m/s and 1.22 m/s, respectively, through the [CH][BE] and [CH][LAC] ILs. The permeability coefficient of H${}_{2}$S is more than 5 times larger than the one for CO${}_{2}$ in both systems, despite its polarity (i.e., permanent dipole moment) and hydrogen-bonding capability. Note that the permeability of both H${}_{2}$S and CO${}_{2}$ through the considered ILs is an order of magnitude larger than their permeability through a lipid bilayer [57,58]. Interestingly, the permeability coefficient of CH${}_{4}$ is 5 orders of magnitude smaller in both ILs. This can be related to the larger size of CH${}_{4}$ as compared to the other molecules. We could not make a direct comparison with experiments as we could not find measured permeability coefficients of these molecules through the considered systems in literature.

To estimate the separating capacity of the ILs, we have defined the selectivity of the considered ILs as the ratios of the permeability coefficients of the corresponding molecules (i.e., permselectivity). The results for the permselectivities are also given in Table 1. Both ILs show exceptionally high CO${}_{2}$/CH${}_{4}$ and H${}_{2}$S/CH${}_{4}$ selectivities as compared to supported ionic liquid membranes (SILMs) (see Reference [59] for a review on SILMs). For example, a maximum CO${}_{2}$/CH${}_{4}$ selectivity of 59 was reported for 1-ethyl-3-methylimidazolium/bis(trifluoromethanesulfonyl)amide ILs on a polymeric membrane support [60], whereas the cholin e-based ILs considered here show more than two orders of magnitude larger CO${}_{2}$/CH${}_{4}$ selectivity. The considered systems are more selective to H${}_{2}$S molecules as compared to CH${}_{4}$ and CO${}_{2}$ molecules (see last two columns in Table 1).

To study the behavior of the considered gas molecules at the gas-to-liquid interface, we have calculated the permeability coefficients by integrating Equation (Equation 1) from |z| = 1.5 nm to |z| = 3 nm (see Figure 2). As presented in Table 2, the permeability coefficients are larger for all three gas molecules through the interface. For CO${}_{2}$ and H${}_{2}$S gas molecules, larger permeability coefficients are obtained in [CH][BE] IL as compared to [CH][LAC] IL. However, the latter IL shows more than 2 time larger permeability for methane. Selectivities of both ILs reduce considerably; they show almost the same selectivity for H${}_{2}$S and CO${}_{2}$ gas molecules. Selectivity of CH${}_{4}$ as compared to the other two gas molecules is also reduced considerably in both materials.

## 4. Conclusions

Using MD simulations, we study the translocation of CO${}_{2}$, H${}_{2}$S and CH${}_{4}$ molecules through choline-based ILs. The free energy and diffusion rate profiles are constructed, which enabled us to calculate the permeability coefficients of the considered molecules. The considered ILs impose the largest resistance to CH${}_{4}$ molecules, whereas 4-5 orders of magnitude larger permeability coefficients are obtained for the other two molecules. Consequently, exceptionally high CO${}_{2}$/CH${}_{4}$ and H${}_{2}$S/CH${}_{4}$ selectivities are predicted for the studied systems. Such selective permeability makes the considered ILs potential candidates for gas separation applications. The obtained results may also trigger further theoretical and experimental research to explore the gas absorption properties of the considered materials.

## Figures and Tables

**Figure 1 molecules-24-02014-f001:**
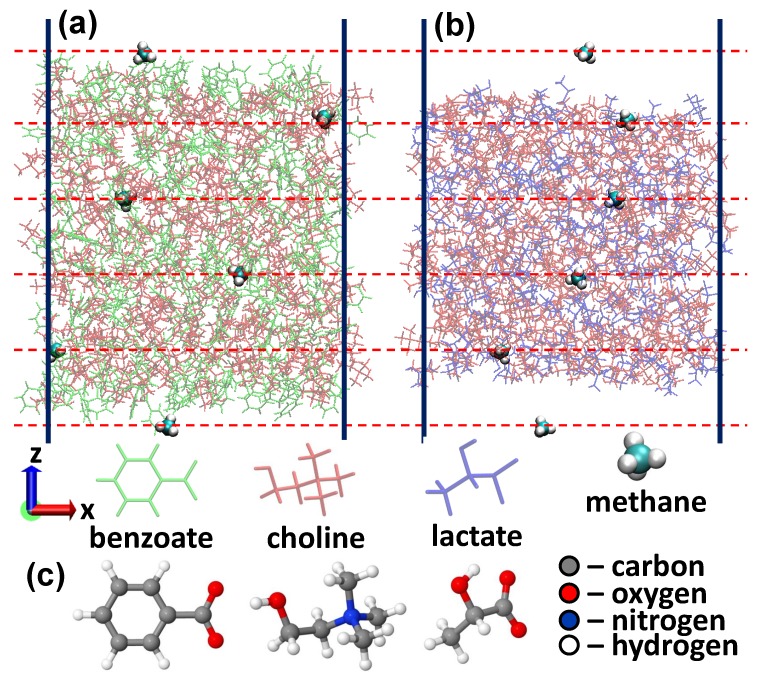
(**a**,**b**) Slabs of choline-benzoate (**a**) and choline-lactate (**b**) ionic liquids with embedded CH${}_{4}$ molecules. (**c**) Optimized structures of benzoate, choline and lactate molecules.

**Figure 2 molecules-24-02014-f002:**
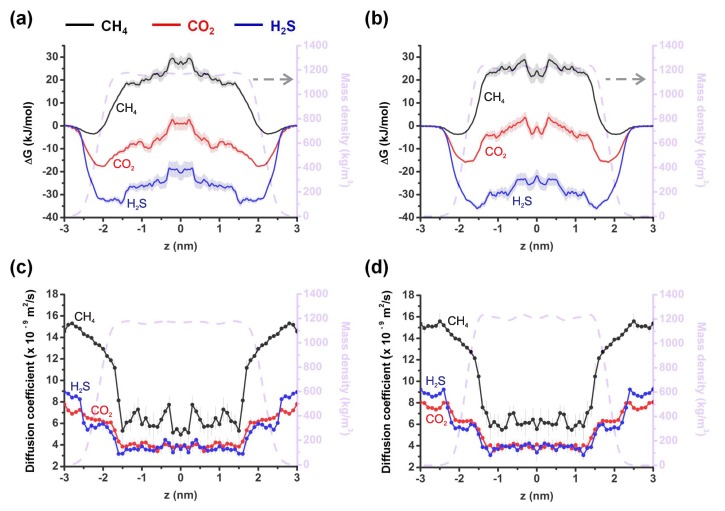
(**a**,**b**) Free energy profiles for the translocation of CH${}_{4}$ (black curves), CO${}_{2}$ (red curves) and H${}_{2}$S (blue curve) across choline-benzoate (**a**) and choline-lactate (**b**) systems. (**c**,**d**) Diffusion coefficient profiles of CH${}_{4}$ (black curves), CO${}_{2}$ (red curves) and H${}_{2}$S (blue curves) in choline-benzoate (**c**) and choline-lactate (**d**) samples. The dashed grey curves in all panels show the mass density of the corresponding systems. Errors associated with the US calculations are depicted in pale color which is obtained by means of bootstrapping method [50].

**Figure 3 molecules-24-02014-f003:**
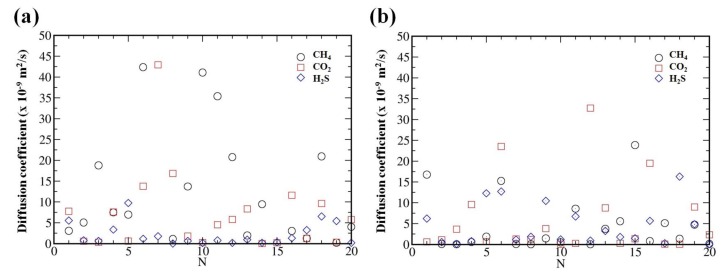
Diffusion coefficients of CH${}_{4}$, CO${}_{2}$ and H${}_{2}$S in choline-benzoate (**a**) and choline-lactate (**b**) samples obtained in 20 various simulations.

**Table 1 molecules-24-02014-t001:** Permeability coefficients of CH${}_{4}$, CO${}_{2}$ and H${}_{2}$S in choline-benzoate and choline-lactate ILs. The last three columns show the CO${}_{2}$/CH${}_{4}$, H${}_{2}$S/CH${}_{4}$ and H${}_{2}$S/CO${}_{2}$ selectivities of the considered ionic liquids defined as the ratios of the permeability coefficients.

	P${}_{{\mathrm{CH}}_{4}}$ (m/s)	P${}_{{\mathrm{CO}}_{2}}$ (m/s)	P${}_{H_{2}S}$ (m/s)	CO${}_{2}$/CH${}_{4}$ Selectivity	H${}_{2}$S/CH${}_{4}$ Selectivity	H${}_{2}$S/CO${}_{2}$ Selectivity
BENZ-CHOL	8.3 × 10${}^{-5}$	2.25	13.21	2.71 × 10${}^{4}$	1.59 × 10${}^{5}$	5.87
LACT-CHOL	7.6 × 10${}^{-5}$	1.22	8.27	1.61 × 10${}^{4}$	1.09 × 10${}^{5}$	6.78

**Table 2 molecules-24-02014-t002:** Permeability coefficients and selectivities obtained by integrating Equation (1) from |z| = 1.5 nm to |z| = 3 nm (see Figure 2).

	P${}_{{\mathrm{CH}}_{4}}$ (m/s)	P${}_{{\mathrm{CO}}_{2}}$ (m/s)	P${}_{H_{2}S}$ (m/s)	CO${}_{2}$/CH${}_{4}$ Selectivity	H${}_{2}$S/CH${}_{4}$ Selectivity	H${}_{2}$S/CO${}_{2}$ Selectivity
BENZ-CHOL	6.8 × 10${}^{-2}$	50.0	53.0	7.31 × 10${}^{2}$	7.75 × 10${}^{2}$	1.06
LACT-CHOL	1.7 × 10${}^{-1}$	28.4	33.1	1.62 × 10${}^{2}$	1.89 × 10${}^{2}$	1.17

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
