# Peer review of "Permeabilities of CO2, H2S and CH4 through Choline-Based Ionic Liquids: Atomistic-Scale Simulations"

_molecules, 2019, doi:10.3390/molecules24102014_

Round 1

Reviewer 1 Report

The authors present a succinct study on the modelling of the permeabilities of CO2, H2S and CH4 in different two different ionic liquids.

I think the work is interesting, however, I have the impression that the material presented is not enough for a full manuscript and should rather be presented as a short communications.

Manuscript should be improved. For example, results and discussion session should start with an introduction sentence, rather than by immediately describing “Figure XX”

Figure 3, it is not clear how can data be taken from this image. Figure 3 shows the different diffusion coefficients for a number of different simulations, however that results show huge differences, which in some cases go beyond 1 order of magnitude. It is not clear what do readers can take form these results. Authors are invited to elaborate on this matter.

It would be interesting to present the selectivity’s determined in comparison to the ones presented in the literature with other IL’s to compare the performance of these systems. How do the authors explain the increase of about two orders of magnitude in selectivity using the choline derived systems? Are is the physical meaning of such high values obtained? Are there any reports in the literature with such high values for selectivity? Authors are invited to elaborate on this matter.

Quality of figure 1 should be greatly improved.

English should be revised:

Gases instead of gasses

3. Results and discussion (not discussions)

Author Response

The authors present a succinct study on the modelling of the permeabilities of CO2, H2S and CH4 in different two different ionic liquids.

Q1. I think the work is interesting, however, I have the impression that the material presented is not enough for a full manuscript and should rather be presented as a short communications. Manuscript should be improved. For example, results and discussion session should start with an introduction sentence, rather than by immediately describing “Figure XX”

A1. Following the comment of the referee, we have added more information both in the introduction part of the manuscript and for the discussions of the results. In the revised manuscript, the changes are highlighted by bold text.

Q2. Figure 3, it is not clear how can data be taken from this image. Figure 3 shows the different diffusion coefficients for a number of different simulations, however that results show huge differences, which in some cases go beyond 1 order of magnitude. It is not clear what do readers can take form these results. Authors are invited to elaborate on this matter.

A2. It is well known that in viscous systems such as ionic liquids it is difficult to obtain dynamic quantities such as self-diffusion coefficients from MD simulations due to, e.g., strong coulombic interactions between the molecular ion constituents. Therefore, one needs to develop special force fields by adopting the existing ones for ILs or create new force fields based on ab initio calculations as is done in the present manuscript. In addition, MD trajectories must be propagated for longer time intervals (ns range) and good statistics should be created by averaging from different initial runs.

Here, we have conducted MD simulations for 50 ns to reach the dynamic equilibrium. The simulations beyond this time become computationally very expensive and can not be performed using available computer simulations. Even for such long MD trajectories, we obtained large fluctuations in determining diffusion coefficients (see Fig. 3) which can be related to the nature of the ionic liquids (e.g., high viscosity due to strong coulombic interactions). Therefore, we have conducted simulations over multiple time origins (i.e., 20 different initial configurations) to improve the statistics.  

In the revised version of the manuscript we have added more text about the possible origin for such large fluctuations. We also mentioned about the large (to our understanding) statistical average in estimating the diffusions coefficients of the considered gas molecules in such viscous media. These calculations also enabled us to compare the results for the diffusion coefficients with the results obtained in umbrella sampling simulations. We hope that the revised text is useful to extract information from the date presented in Fig. 3.

Q3. It would be interesting to present the selectivity’s determined in comparison to the ones presented in the literature with other IL’s to compare the performance of these systems. How do the authors explain the increase of about two orders of magnitude in selectivity using the choline derived systems? Are is the physical meaning of such high values obtained? Are there any reports in the literature with such high values for selectivity? Authors are invited to elaborate on this matter.

A3. In their recent review paper, Karkhanechi et al., have presented CO2/CH4 selectivities of several ionic liquids [ChemBioEng Rev 2015, 2, 290-302]. Our predicted selectivities are indeed larger than the ones presented in this paper. To our understanding, such larger values for the selectivity could be due to the finite size effects in our systems. Limited simulations time could also affect the obtained results. In the revised version of the manuscript, we have presented the selectivity values determined from the permeability coefficients determined at the gas-liquid interfaces (see Table II in the revised manuscript). The selectivity values drops by more than two orders of magnitude due to larger permeability values for the methane.   

Q4. Quality of figure 1 should be greatly improved.

A4. We have improved the quality of this figure in the revised version of the manuscript.

Q5.English should be revised:

Gases instead of gasses

3. Results and discussion (not discussions)

A5. We thank the referee for his/her careful reading of the manuscript. We have our best to improve the English of the manuscript.

Reviewer 2 Report

In this article, the authors applied molecular dynamics simulations to calculate permeability coefficients of CO2, H2S, and CH4 in choline-benzoate and choline-lactate ionic liquids. The GROMOS force field parameters for all molecules were obtained from the automated topology builder (ATB) repository based on DFT calculations. The authors should discuss this parametrization in more details, especially their choice of charges for ions of ionic liquids. ATB repository lists benzoate, lactate, and choline with net charge +1/-1. The transport properties of ionic liquids are not well described when full +1/–1 charges are assigned to the cations and anions (see for instance Youngs et al. ChemPhysChem 2008, 9, 1548 and Zang et al. J. Phys. Chem. B, 2012, 116, 10036). A simple approach used in many MD simulations of ionic liquids is to scale the charges by a factor of 0.7–0.9. Such scaling is important for self-diffusivity of ionic liquids and the authors should discuss why they did not use it and how it could affect their calculations of permeability coefficients.

The authors should also discuss their choice of parameters for free energy profiles calculations. The use of only 6 windows in umbrella sampling simulations separated by 1 nm with quite large force constant of 2000 kJ/mol/nm can give too small overlap between histograms from all windows in the weighted histogram analysis method and large errors in the free energy profiles. I would recommend plotting these histograms. Figure 2 shows quite small errors obtained from bootstrapping method but on the other hand, the free energy profiles show large oscylations close to the boundaries between umbrella sampling windows.

How the average values of the excess free energy were calculated?

There are no details of the method used for analysis of diffusion coefficient of solutes in the bulk ionic liquid shown in Figure 3. Was any optimization performed for randomly placed molecules before starting MD simulations? How possible clashes were avoided? Is equation 2 used to calculate the diffusion coefficient also for solutes in the bulk ionic liquid? For unbiased solutes, diffusion coefficients could be calculated from mean square displacements. There are large errors shown in Figure 3, but are the average values for the diffusion coefficient of solutes in the bulk in agreement with that from diffusion coefficient profiles?

Author Response

Q1. In this article, the authors applied molecular dynamics simulations to calculate permeability coefficients of CO2, H2S, and CH4 in choline-benzoate and choline-lactate ionic liquids. The GROMOS force field parameters for all molecules were obtained from the automated topology builder (ATB) repository based on DFT calculations. The authors should discuss this parametrization in more details, especially their choice of charges for ions of ionic liquids. ATB repository lists benzoate, lactate, and choline with net charge +1/-1. The transport properties of ionic liquids are not well described when full +1/–1 charges are assigned to the cations and anions (see for instance Youngs et al. ChemPhysChem 2008, 9, 1548 and Zang et al. J. Phys. Chem. B, 2012, 116, 10036). A simple approach used in many MD simulations of ionic liquids is to scale the charges by a factor of 0.7–0.9. Such scaling is important for self-diffusivity of ionic liquids and the authors should discuss why they did not use it and how it could affect their calculations of permeability coefficients.

A1. We thank the referee for his/her useful comment about the effect of charge partitioning. Indeed, description of both static and dynamics properties of ionic liquids depend on the charges assigned to the components of the ionic liquids. For example, the full charge approach gives better prediction to the static properties of the ionic liquids, whereas scaled charges give better description for the dynamic properties of the ionic liquids, such as self-diffusivity.  Since static properties (i.e., density of the liquids) of the considered systems already exist in the literature, we decided to do full charge calculations in order to validate our force fields by direct comparison with the experiment.  Indeed, our calculations give good agreement with the experiment for the density of the considered ionic liquids. Since we did not conduct DFT-based charge partitioning calculations (due to high computational cost of DFT simulations for such large systems) we did not use charge scaling method mentioned by the referee. In addition, we were not sure about the consequences of such direct charge scaling on the obtained results. In most of our previous works (see e.g., Phys. Rev. B 91, 014304 (2015); J. Appl. Phys. 118, 025101 (2015); J. Appl. Phys. 120, 225108 (2016); Appl. Surf. Scienc. 416, 725-730 (2017)) we used reactive force fields (ReaxFF) where charges are recalculated in each MD time step in order to account for dynamic bond formation/dissociation. We initially used this force field to describe our systems. However, we could not propagate MD simulations for required time interval due to larger computational cost of reactive calculations for ionic liquids. 

During the resubmission process, we started test calculations for CH4 and CO2 gas molecules by scaling the changes of the [CH][LAC] ionic liquid components by 80% (due to limited access to the HPC facilities of UA, we could not complete the calculations for the scaled charges). We have obtained an increase in the diffusion coefficients of both molecules, especially in the bulk of the material. However, similar rate of enhancement is obtained for the diffusion of both molecules. Therefore, we expect no qualitative changes in the selectivity of the considered ionic liquids for the considered gas molecules.

In the revised version of the manuscript, we have mentioned about the effect of charge partitioning on the dynamic properties of the ionic liquids.

Q2. The authors should also discuss their choice of parameters for free energy profiles calculations. The use of only 6 windows in umbrella sampling simulations separated by 1 nm with quite large force constant of 2000 kJ/mol/nm can give too small overlap between histograms from all windows in the weighted histogram analysis method and large errors in the free energy profiles. I would recommend plotting these histograms. Figure 2 shows quite small errors obtained from bootstrapping method but on the other hand, the free energy profiles show large oscillations close to the boundaries between umbrella sampling windows. How the average values of the excess free energy were calculated?

A2. We used 10 Å spacing between each window in umbrella sampling knowing that such a separation is commonly used in studying ion transport through membranes. Convergence is checked after certain time run for each window. In addition, finale energy profiles are obtained using more than 250 umbrella sampling simulations.

As for the force constant, to the best of our knowledge, there is no defined rule and the choice depends on the system parameters and the type of simulations. We have chosen such larger value for the force constant due to very strong intermolecular interactions in ionic liquids. Unfortunately, we could not complete the simulations for the other values of the force constant due to limited computational resources.   

Q3. There are no details of the method used for analysis of diffusion coefficient of solutes in the bulk ionic liquid shown in Figure 3. Was any optimization performed for randomly placed molecules before starting MD simulations? How possible clashes were avoided? Is equation 2 used to calculate the diffusion coefficient also for solutes in the bulk ionic liquid? For unbiased solutes, diffusion coefficients could be calculated from mean square displacements. There are large errors shown in Figure 3, but are the average values for the diffusion coefficient of solutes in the bulk in agreement with that from diffusion coefficient profiles?

A3. To avoid artificial problems in the calculations due to the overlap between the atoms of the ionic liquids and newly created gas molecules, we have used large enough intermolecular distance during the generation of the gas molecules using Packmol. In addition, we have conducted structural optimizations for every random generation of the gas molecules inside the system and conducted NPT simulations to optimize lattice parameters of the simulation box before running long NVT simulations. The diffusion coefficients are obtained from the mean square displacements using Einstein relation. The averaged values of the diffusion coefficients obtained in these calculations are in good agreement with the values from umbrella sampling methods. This is mentioned in the revised version of the manuscript.   

We thank the referee for his/her useful comments/suggestions.

Reviewer 3 Report

Amhamed and coauthors present the results of molecular dynamics simulations examining the ability of benzoate-choline and lactate-choline ionic liquids to sequester CO2, H2S and CH4. This was done by calculating the permeability of each species through both ILs. The results are generally compelling and well-described. I have some questions and concerns, but they should be addressable by the authors with revisions.

Major Points:

1.       What are ionic liquids? Why are ionic liquids appropriate for this application?

2.       Why are benzoate-choline and lactate-choline ionic liquid chosen?

3.       Are the free energy profiles symmetrized?

4.       What are ‘permeability coefficients’? What are values for other species in ionic liquids?

5.       Figure 3: The sporadic behavior observed in Fig. 3 makes me wonder how the authors are sure that the simulations are long enough to be converged and fully representative of equilibrium behavior of these gases in the ILs. Particularly diffusion, which is a very slowly converging property, particularly in a viscous fluid like and IL. What are the viscosities of these ILs?

6.       Line 119: What do the authors mean by “hopping mechanism”? Can this we explored further using a residence time or some kind of clustering analysis?

Minor Points:

7.       Line 21 “present [the] possibility”

8.       Line 56: “...Packmol package…” -> “…the Packmol package…”

9.       Line 57: Should the second “and” be “or”?

10.   Line 67: The “e” in Nose should have an acute accent (that is, é).

11.   Line 98: Maybe permanent dipole moment would be better than “finite dipole moment”.

12.   Fig. 2: The dashed grey is very difficult to see. Maybe a darker grey, or even just green, could be used instead?

13.   Line 139  “..more than two orders of [magnitude] larger”

Author Response

Amhamed and coauthors present the results of molecular dynamics simulations examining the ability of benzoate-choline and lactate-choline ionic liquids to sequester CO2, H2S and CH4. This was done by calculating the permeability of each species through both ILs. The results are generally compelling and well-described.

We thank the referee for his/her positive opinion about our manuscript.

Q1. I have some questions and concerns, but they should be addressable by the authors with revisions.

Major Points:

What are ionic liquids? Why are ionic liquids appropriate for this application?

A1. Ionic liquids consist of large organic cations and organic or inorganic anions with low melting points. They exhibit unique physical and chemical properties such as good thermal properties, improved structural stability, non-flammability, and more interestingly very low vapor pressure. Ionic liquids have a potential for gas separation applications due to small degradation, high absorption rate and environmental issues. The operational cost of ionic liquids for such applications is much smaller than the one for conventional materials. Moreover, there is possibility of creating task-specific ionic liquids with desired properties.

The revised version of the manuscript contains the above information.

Q2.       Why are benzoate-choline and lactate-choline ionic liquid chosen?

A2. Choline-cation-based ionic liquids have recently attracted considerable attention due to required environmental, toxicological, and economical profiles for practical applications in many different technological areas, such as acid gas removal. Most importantly, these ionic liquids can be synthesized using low cost procedures. This information is already included in the manuscript.   

Q3.       Are the free energy profiles symmetrized?

A3. Yes, energy profiles are symmetrized.

Q4. What are ‘permeability coefficients’? What are values for other species in ionic liquids?

A4. Permeability coefficient (or permeability) is an experimental measure of the transport flux of the material through a membrane per unit driving force per unit thickness. There are several methods of defining permeability in computer simulations. In this work the permeability is calculated using spatially dependent diffusion coefficients. The review paper by Karkhanechi et al. [ChemBioEng Rev 2015, 2, 290-302] summarizes the permeability coefficients of the other ionic liquids (see Ref. [59] in the manuscript). 

 Q5. Figure 3: The sporadic behavior observed in Fig. 3 makes me wonder how the authors are sure that the simulations are long enough to be converged and fully representative of equilibrium behavior of these gases in the ILs. Particularly diffusion, which is a very slowly converging property, particularly in a viscous fluid like and IL. What are the viscosities of these ILs?

A5. Most of the ionic liquids are characterized by very high viscosities originating from very strong coulombic interactions between the molecular ion constituents. Therefore, MD trajectories must be propagated for longer time intervals (ns range) and good statistics should be created by averaging from different initial runs. In our simulations, we propagated the gas molecules for 50 ns, which, to our knowledge, is long enough to get to the diffusive regime. In addition, we have conducted statistical analysis by sampling over 20 different random locations of the gas molecules.  The average values of the diffusion coefficients obtained from the mean square displacements are in good agreement with the ones obtained in umbrella sampling calculations. This information is given in the present version of the manuscript.

Q6.       Line 119: What do the authors mean by “hopping mechanism”? Can this we explored further using a residence time or some kind of clustering analysis?

A6. After careful analysis of the date, we decided to remove this statement from the manuscript due to lack of support for our claim.

Q7. Minor Points:

7.       Line 21 “present [the] possibility”

8.       Line 56: “...Packmol package…” -> “…the Packmol package…”

9.       Line 57: Should the second “and” be “or”?

10.   Line 67: The “e” in Nose should have an acute accent (that is, é).

11.   Line 98: Maybe permanent dipole moment would be better than “finite dipole moment”.

12.   Fig. 2: The dashed grey is very difficult to see. Maybe a darker grey, or even just green, could be used instead?

13.   Line 139  “..more than two orders of [magnitude] larger”

A7. We thank the referee for his/her carefully reading of our manuscript. We have included corresponding changes into the revised version of the manuscript.

We thank the referee for  his/her positive opinion about our work and for useful comments/suggestions.

Reviewer 4 Report

The MS is interesting and can be published after minor revision, according to the following:

1) In the Introduction, the Authors should specify the motivations of their choice of using B3LYP/6-31G* instead of other similar approaches. 

2) As far I can read from the materials at my disposal, I could not find any supplementary materials reporting Cartesian coordinates and total energies for all calculated structures. Those data should, in my opinion, be provided in some form from the Authors.

Author Response

The MS is interesting and can be published after minor revision, according to the following:

Q1. 1) In the Introduction, the Authors should specify the motivations of their choice of using B3LYP/6-31G* instead of other similar approaches. 

A1. Since we could not find existing classical force fiends for the considered ionic liquids, we referred to the automated topology builder (ATB) repository to generate the necessary force fields. By default ATB uses B3LYP/6-31G* level of theory + Hessian to create force fields for the molecules containing less than 40 atoms, which is the case in the present work. This is now mentioned in the introductory part of the revised version of the manuscript.      

Q2. 2) As far I can read from the materials at my disposal, I could not find any supplementary materials reporting Cartesian coordinates and total energies for all calculated structures. Those data should, in my opinion, be provided in some form from the Authors.

Q2. During the resubmission processes we present XYZ/CIF files of both systems studied in the present work.

We thank the referee for his/her positive opinion about the manuscript and for useful comments.

List of the changes:

1.     Page 1, column 1: Third sentence from the bottom is added.

2.     Page 1, column 2, paragraph 3: Sentence 2 is added.

3.     Page 1, column 2: Last sentence is added.

4.     Page 2: The quality of Fig. 1 is improved.

5.     Page 2, column 1: first two sentences are added.

6.     Page 2, column 2: Note about supplemental material is added.

7.     Page 4, column 1, paragraph 1: first two sentences are added.

8.     Page 4, column 1, paragraph 2: first two sentences are added.

9.     Page 4, column 2: First 2 paragraphs are added.

10.  Page 5: Table 2 is added.

11.  Page 5, column 2: First paragraph is added.

12.  References 53-56 are added.

Round 2

Reviewer 1 Report

Authors have addressed the comments and the manuscript is now suitable for publication.

Reviewer 2 Report

The authors have answered all my questions and have revised the manuscript accordingly. I recommend to accept the manuscript in the present form.